# OpenReview forum: "On High-Dimensional Action Selection for Deep Reinforcement Learning"
_ICLR.cc/2025/Conference — ICLR 2025 Conference Withdrawn Submission_

### Official Review · Reviewer_z2C1 · 2024-10-18

**Soundness:** 2
**Presentation:** 3
**Contribution:** 2
**Rating:** 3
**Confidence:** 4

**Summary:**

The authors propose a new method for reducing the action space dimension in online RL scenarios. For that, they leverage the model-X knockoff method. In short, the authors propose to construct a set of impactful action dimensions that is updated a few times during training. They select the action dimensions by resampling an action after each environment interaction and evaluate how much each action dimension is useful for predicting the reward and next state compared to the resampled action. The action dimensions with the greatest difference are considered meaningful.

**Strengths:**

1. The paper investigates an important topic in RL.

2. The presented algorithm performs strongly against the proposed baseline in the considered experiments.

**Weaknesses:**

1. In its current form, many crucial explanations are missing, which drives down the quality of the submission:

     a. The motivating example in Figure 1 could be more detailed. Many details are missing, such as the environment, the number of seeds, etc.

     b. In Equation 2, it is unclear if the set over which the minimum is taken always contains an element. If this set is empty, $\tau_{\alpha}$ would not be properly defined. Is there any condition on the $W_j$ to ensure that $\tau_{\alpha}$ is properly defined? For example, assuming $\alpha < 1$, if $W_0 = -1, W_1 = -0.5$, and $W_2 = 0.5$, the set is empty as the ratio is always greater than 1.

     c. In Figure 2, the authors include a variable selection method without citing it. Which method is used in this experiment? The authors should also briefly explain the idea behind this method.

     d. As opposed to the authors claim in Line 464, the method seems to require additional time as each action is sampled twice and Algorithm 2 consist on a loop over the entire replay buffer. In order to fairly compare the methods, could the authors report the same performance plots with the clock time as $x$-axis instead of the number of gradient steps? Even if time efficiency is not the main objective of the paper, I believe this analysis would help clarify the benefits and drawbacks of the presented method.

     e. In Line 344, the authors claim that the trajectories can be assumed independent without loss of generality. Could the authors add a justification?

     f. In Line 358, the authors do not explain why the flip sign property needs to be satisfied in knockoff-type methods.

     g. The proof of Theorem $4.3$ is not entirely written in the appendix. The authors indirectly ask the reader to make the proof by referring to another work. I argue that the proof is needed and that the paper cannot be accepted without further explanation.

     h. Section $4$, ends abruptly without explaining why Theorem $4.3$ is relevant for the presented method. Including an explanatory paragraph would improve the significance of the contribution.

     i. The baselines introduced in Line 425 are not explained. Similarly, the analysis of Figure 5 could be further developed as, in its current form, the reader is left with little information.

     j. See Questions.

2. The experimental results could be further elaborated:

     a. It would be interesting to analyze the behavior of the proposed method for more complex environments of the MuJoCo benchmark.

     b. An ablation study on the type of machine learning algorithm used to predict the reward and the next state would strengthen the authors' analysis.

     c. Although many baselines are mentioned in the Related Work, the authors do not compare their method against any other baselines except the one using the full action space for the MuJoCo environments. Can the authors comment on this decision?

**Questions:**

1. In their experiments, the authors use LASSO to predict the reward or a dimension of the next state. The reward and transition dynamics are, in most cases, non-linear. Could the authors add a discussion on the impact of this non-linearity?

2. The number of times the subset of action dimension is recomputed is not mentioned. Specifically, the value of $T_{vs}$ is not shared. What is the chosen value? Has this value an impact on the performances?

3. Could the authors reveal the choice of the anti-symmetric function $f$ used for the experiments?

4. The action distributions in Figure C.1 are looking alike. Could the authors comment on this aspect?

– Remarks –

A. Line 142, it is not clear what $\rho_{\pi}(s_t)$ and $\rho_{\pi}(s_t, a_t)$ stand for as $s_t$ and $a_t$ are not defined. I suggest defining $\rho_{\pi}^t(s)$ as the probability of being in state $s$ after $t$ steps when the policy $\pi$ is used.

B. In definition 2.1, $S_t, A_t, R_t$ and $S_{t+1}$ are not defined. A definition should be self-contained.

C. An interpretation of the $\alpha$ parameter in Equation 2 would be useful to help understand its significance.

D. In Algorithm 2, the index $j$ is used to index the action dimension, whereas, in Line 311, $j$ is used to index the trajectories. I suggest using another letter to index the trajectories.

E. Figure 2 (left) legend indicates PPO instead of SAC.

F. Line 332, the authors refer to Section $4$ instead of Section $2.3$. Furthermore, “equation” should take a capital letter.

G. I suggest explaining the authors' intentions in Section $4$ before diving into the content. This would help the reader dive in the section.

H. The font of all the figures should be increased. In Figure 3, a lot of space is lost between the plots, which could be used to increase the plots' size.

---

### Official Review · Reviewer_DQZh · 2024-10-29

**Soundness:** 2
**Presentation:** 2
**Contribution:** 1
**Rating:** 3
**Confidence:** 2

**Summary:**

This paper presents a deep reinforcement learning (RL) framework that incorporates Knockoff sampling to handle high-dimensional action spaces. In the problem setting examined, actions consist of dimensions that influence state transitions and dimensions that do not. The proposed algorithm identifies the informative action dimensions using Knockoff sampling and masks actions based on the selected dimensions.

The method was evaluated on continuous control tasks in MuJoCo, where the action space was augmented with additional dimensions that have no effect on state transitions. Experimental results demonstrate that the proposed method effectively identified the influential action dimensions and achieved performance close to that obtained when the effective dimensions were known.

**Strengths:**

- The proposed method improves the performance of the original PPO and SAC on tasks with action spaces augmented by dimensions that do not affect state transitions.

- The presentation is overall clear.

**Weaknesses:**

- The problem setting considered in this study does not appear practical. I cannot think of real-world tasks where the action space is augmented with meaningless dimensions. For instance, in controlling humanoid robots, the action space is high-dimensional, but all dimensions impact state transitions.

- The novelty of the proposed method seems limited. If I understand correctly, the algorithm works as follows: (1) perform knockoff sampling to select action dimensions, (2) mask the action to exclude meaningless dimensions, and (3) apply an off-the-shelf RL algorithm. Since the proposed algorithm merely combines knockoff sampling with RL, I do not see a substantial novelty in the algorithm.

**Questions:**

- If the authors have specific real-world problems in mind that the proposed method could address, please share them.

- If possible, please evaluate the proposed method on more realistic tasks that are applicable to real-world scenarios.

---

### Official Review · Reviewer_6j4o · 2024-11-03

**Soundness:** 3
**Presentation:** 2
**Contribution:** 3
**Rating:** 8
**Confidence:** 3

**Summary:**

This paper proposes a method called Knockoff Sampling (KS) for online RL, designed to identify a minimal sufficient action set in high-dimensional action spaces. Knockoff sampling generates knockoff features from the policy, and to apply this to RL, the authors introduced an action mask for the policy and value function, sample splitting to reduce temporal dependency, and majority voting to integrate the split results. Experiments in MuJoCo and Treatment Allocation tasks show that the KS method effectively selects relevant actions and, when combined with RL algorithms, achieves performance comparable to learning with only the actual relevant actions.

**Strengths:**

1. Selecting essential actions in large action dimensions and learning RL based on these is a crucial and necessary area of research. The Knockoff Sampling (KS) method proposed in this paper is expected to make a significant contribution in this field.
2. The approach builds on the existing model-X knockoff method by adding an action mask, sample splitting, and majority voting, which have been successfully adapted for RL applications.
3. KS is theoretically proven to not only select the minimal sufficient action but also control the False Discovery Rate (FDR).
4. When applied to RL, the proposed KS method achieved performance similar to using only the actual relevant actions, and experiments demonstrated that KS effectively identifies the truly relevant actions.

**Weaknesses:**

1. While there is no issue with the definition of the sufficient action set, it seems overly strict. In realistic scenarios, especially with high action dimensions, each action dimension may influence the state without necessarily impacting task completion. A more flexible definition of the sufficient action set would be beneficial.
2. The variable selection method itself appears to rely on the existing model-X knockoffs approach. Proposing a novel selection method in this aspect could have added more originality to the research.
3. The experimental setup could be improved. Adding dummy action dimensions in the MuJoCo environment seems distant from real-world scenarios and may be overly simplistic.
4. Details regarding the Treatment Allocation Task are lacking. Since it was referred to as a dataset, it appears to be a non-RL environment, making the choice to use an online algorithm like PPO appear unusual.
5. It would be beneficial to include comparisons between KS, true action, all action, and VS within the MuJoCo environment. Additionally, demonstrating that VS has a high False Discovery Rate (FDR) would strengthen the evaluation and provide a clearer picture of KS’s advantages.

**Questions:**

1. Is the Treatment Allocation Task an RL environment? Using an online and on-policy RL algorithm like PPO with only a dataset seems unusual. Could you provide more details about this task?
2. What was the rationale for selecting Lattice and gSDE as comparison algorithms in the Treatment Allocation Task? Since these algorithms lack references in the related work section, a brief explanation of each algorithm and the reason for choosing them as baselines would be helpful.
3. Could you provide comparisons of return and FDR between KS, true action, all action, and VS in the MuJoCo environment?

---

### Official Review · Reviewer_frxR · 2024-11-03

**Soundness:** 2
**Presentation:** 3
**Contribution:** 3
**Rating:** 5
**Confidence:** 4

**Summary:**

The paper addresses the problem of high-dimensional action selection in deep RL, where redundant actions can slow down learning and increase computational demands. To tackle this, the authors propose a data-driven knockoff sampling (KS) approach that dynamically identifies a minimal set of essential actions during training, improving efficiency without requiring prior domain knowledge. The method integrates KS into online RL with a mask mechanism that prunes non-essential actions, effectively reducing the dimensionality of the input space. The authors also provide theoretical guarantees that KS controls the false discovery rate even with the dependent data typical in RL. Empirical results in MuJoCo environments and a sepsis treatment allocation task demonstrate that the KS approach enhances learning efficiency and achieves rewards close to those obtained by models with pre-known essential actions, outperforming models trained on the full action space.

**Strengths:**

1. Important problem in high-dimensional action selection: The paper addresses a significant challenge in RL—efficiently selecting actions in high-dimensional spaces. In fields like robotics and healthcare, RL applications often involve large action spaces with many redundant or minimally influential actions that can slow down learning and increase computational demands.

2. Adaptation of Knockoff Sampling to RL: The paper creatively adapts knockoff sampling, a feature selection method, for identifying essential actions in RL. By integrating a novel masking mechanism, it dynamically prunes irrelevant actions during training, effectively reducing dimensionality without requiring domain-specific knowledge.

3. Sound methodology with theoretical guarantees: The method is robustly designed with theoretical guarantees for controlling the false discovery rate in RL with temporally dependent data, using sample splitting and majority voting. These adaptations add reliability, particularly under the stationary environment and independent action assumptions, supporting its rigor within the defined scope.

4. Clarity and detailed explanation: The paper is well-organized and clearly explains the knockoff sampling adaptation and the action-masking mechanism, making the technical contributions accessible.

5. Promising step toward scalable RL: Although limited to stationary environments and independent actions, the approach provides a valuable step toward scalable RL in high-dimensional settings.

**Weaknesses:**

1. Overly simplistic assumptions on action distributions: The paper assumes a multivariate Gaussian distribution with independent actions, a strong simplification that doesn’t reflect the complexity of most real-world RL tasks, where actions are often interdependent. Although the authors mention that the method could extend to correlated actions, this extension is neither theoretically developed nor tested empirically. The assumptions limit the approach’s applicability to only a subset of RL problems and leave open questions about its performance in more realistic scenarios with action dependencies.

2. Stationary environment assumption: The method is developed under the assumption of stationary environments, which is not representative of many RL applications where dynamics evolve over time. This limitation could affect the generality of the proposed solution, as adapting knockoff sampling and FDR control to non-stationary contexts could require substantial modifications. Testing the method in at least partially non-stationary environments would provide a stronger validation.

3. Simplified experimental setup: In the experiments, the authors artificially add redundant actions that are completely independent of the environment’s dynamics and rewards. This setup makes the action selection task artificially easy, as the redundant actions are trivially irrelevant. In realistic settings, irrelevant actions may still have weak or noisy influences on rewards and state transitions, complicating the task of distinguishing essential from non-essential actions. Testing the method in environments where redundant actions are interdependent or only partially relevant would better demonstrate the robustness and practical applicability of the approach.

4. Limited empirical evaluation of FDR control in RL: Although the authors provide theoretical guarantees for FDR control, the empirical validation is limited to specific synthetic environments with idealized conditions. FDR control in RL is a relatively new and challenging area, and it would strengthen the paper to include more diverse environments or scenarios where FDR control is explicitly evaluated and shown to hold under more realistic settings.

5. Unexplored impact of masking on learning dynamics: While the masking mechanism reduces dimensionality, the paper does not fully explore how this pruning affects the learning process or convergence. In RL, sudden changes in the action space could potentially destabilize training or lead to suboptimal exploration strategies. An analysis of the impact of masking on the RL agent’s exploration behavior and convergence speed would provide useful insights and improve the reliability of the method.

6. Lack of comparison with existing baselines: The paper would benefit from comparisons with established methods for managing high-dimensional action spaces in RL, such as Growing Action Spaces (GAS) and Action Elimination (AE-DQN) amongst others. GAS incrementally expands the action space over training, allowing agents to explore efficiently by starting with a restricted subset of actions and progressively increasing complexity. AE-DQN, on the other hand, eliminates actions deemed irrelevant in specific states, helping to reduce exploration time and computational load by focusing on potentially optimal actions. Adding comparisons with these baselines would give a clearer picture of how knockoff sampling and masking stack up against other dimensionality reduction and action selection techniques, helping to highlight both the advantages and any limitations of the proposed approach.

7. Lack of state-dependent action pruning: The method applies a fixed action mask that does not adapt based on the current state, limiting its flexibility in environments where action relevance varies across states. This global, static pruning contrasts with state-dependent methods (e.g. Action Elimination), which adjust the action space dynamically in response to state-specific needs.

**Questions:**

1. Testing with more realistic actions: Given that the added actions are fully independent of the main action set, the current results may not fully reflect the method's capability in more realistic settings. Could you add experiments with partially correlated redundant actions to better simulate real-world challenges where irrelevant actions might still have weak or moderate influence? One way to introduce these correlations may be to create additional actions as noisy or linear combinations of essential actions, introducing varying levels of correlation.

2. Empirical validation of FDR control in varied RL settings: FDR control is validated only in limited, controlled environments with idealized conditions. Could you conduct further tests of FDR control in more complex RL environments, especially those with action dependencies or larger state-action spaces? Tracking false positives and false negatives across different settings, and evaluating the stability of FDR over time, may provide practical insights.

3. Comparison with established baselines: Since comparisons with existing methods including Growing Action Spaces and Action Elimination are crucial to evaluating relative strengths and weaknesses, could you add these baselines to situate the knockoff sampling method among existing techniques? Was there any specific reason why you only included the "true actions" in your MoJoCo experiments?

4. State-dependent action masking: The current approach applies a static mask across all states, but state-dependent action pruning could improve adaptability in complex environments. Could you discuss any potential adaptations for state-dependent masking?

5. Adaptation to non-stationary environments: The method’s theoretical basis and experiments assume a stationary environment, limiting applicability in dynamic settings. Could you test or discuss potential modifications for non-stationary environments, where action relevance may shift over time?

6. Impact of masking on learning dynamics: Could you add an analysis of how masking affects learning dynamics, particularly exploration and convergence? To gain a better understanding of how masking influences learning stability, you may want to track exploration metrics (e.g., action diversity or visit counts across states) before and after masking is applied, as well as measure convergence rates or instability indicators (e.g., variance in policy updates) following the introduction of masking. This analysis would clarify whether sudden reductions in action space impact the balance between exploration and exploitation or introduce any learning instability.

---

### Note · Authors · 2024-11-25

I have read and agree with the venue's withdrawal policy on behalf of myself and my co-authors.